# Should Glaciers Be Considered Permafrost?

**Maciej Dąbski** 

Faculty of Geography and Regional Studies, University of Warsaw, 00-927 Warsaw, Poland;
mfdabski@uw.edu.pl; Tel.: +48-22-55-20-631

**Abstract:** This commentary critically evaluates concepts of extending the term permafrost to any parts of an active glacier. The whole mass of any glacier is at zero centigrade or below (cryotic), except for non-ice inclusions at the glacier surface. Therefore, if glacial ice is considered a monomineral rock, then any glacier constitutes a perennially cryotic ground (i.e., permafrost), according to the purely thermal definition. However, extending the term permafrost to active glaciers introduces misconceptions, rather than a clarification of important geological terms.

**Keywords:** glacial permafrost; permafrost; glacier–permafrost interactions; polythermal glacier

## 1. Introduction

The concept of "glacial permafrost" was first introduced by Hughes [1] in 1973 to emphasise the role of glacier dead ice blocks buried in permafrost. Recently, Dobiński [2–4] and Dobiński et al. [5,6] discussed our understanding of the relationships between glacial and periglacial domains and argued that the term glacial permafrost should be used in addressing cold glacier ice in active glaciers as a continuation of proglacial permafrost. The aim of this commentary is to show weaknesses of the aforementioned concept and to advocate for exclusion of any active glaciers (temperate, polythermal or cold) or their parts from the term permafrost.

## 2. Permafrost and Glacial Permafrost

According to the International Permafrost Association (IPA), permafrost is defined as a ground remaining at or below 0 °C for at least two consecutive years [7,8]. According to this definition, dry sand or solid massive rocks with no ice are also permafrost as long as they remain perennially cryotic. Dobiński draws from this definition far-reaching conclusions which brought him to promote the idea of glacial permafrost, where glacial ice is treated as permafrost. Dobiński [2] argues that the glacial ice, due to its crystalline structure and natural occurrence, should be treated as a perennially cryotic monomineral rock with morainic inclusions, that is, as permafrost. Moreover, a subglacial mineral layer immediately beneath a temperate glacier would also be regarded as permafrost, because it is saturated with cryotic water, not frozen due to high pressure [2].

The Cryosphere Glossary of the National Snow and Ice Data Center (NSIDC) defines permafrost in a more traditional way, as "a layer of soil or rock, at some depth beneath the surface, in which the temperature has been continuously below 0 °C for at least several years; it exists where summer heating fails to reach the base of the layer of frozen ground" [9]. This definition implies the state of freezing, which in turn implies the presence of ice. According to this definition, cryotic but dry sand (e.g., dune) should not be regarded as permafrost. Moreover, the second part of this definition excludes active glacier ice, allowing the application of the term "permafrost" for glacial environments only in a situation of buried dead ice or marginal parts of a glacier which are overlain by a non-ice active layer, for example, supraglacial moraine of fluvioglacial cover (a situation frequently occurring in nature). Therefore, the NSIDC's definition of permafrost differs from the one advocated by Dobiński [2].

The two definitions do not contradict each other in the understanding of what is a rock or a ground, but rather in that a glacier remains frozen all the time (allowing only for partial melting) and no active layer can develop on the glacier.

If we give a glacier the status of a rock, because of its crystalline structure and natural occurrence, we should also regard a snow patch as a rock (of very porous texture), and a perennial snow patch (lasting for at least two consecutive years) should be regarded as permafrost. However, this would be against common sense and would lead only to misconceptions.

## 3. Models of Periglacial and Glacial Permafrost

Dobiński [2] and Dobiński et al. [5] developed a model of continuity between typical "periglacial permafrost" and "glacial permafrost" (or "glacial ice permafrost"), which actually clashes with the notion that all glacial ice is permafrost. This is because in the model only cold glacial ice is treated as permafrost—a continuity of "periglacial permafrost". It could be justified, but only if we apply the term cryotic in a peculiar way when referring to "glacial permafrost". This peculiarity would mean exclusion of temperate glacial ice from the cryotic state. However, the scientific way of reasoning implies strict understanding of definitions of important terms, and understanding of what is cryotic should not be subject of a debate.

Dobiński, in the first mentioned publication [2], drew a cryofront (0 °C isoline) through the interior of a polythermal glacier, separating upper cold ice from lower temperate ice. This is hard to accept, because a glacier is cryotic in its whole volume, with no real cryofront inside. According to the Cryosphere Glossary [9], a cryofront is the boundary between cryotic and non-cryotic ground as indicated by the position of the 0 °C isotherm in the ground. Temperate glacial ice is still cryotic, despite partial melting, because its temperature remains at the pressure melting point, which is 0 °C or lower.

In the second mentioned publication of Dobiński [5], the same drawing is already improved, because the boundary inside a glacier changed its name to "melting point surface" (MPS). This can be accepted only partially, because temperate ice undergoes partial melting in its total volume, and not along a plane which can be illustrated as a single line in a 2D picture. If we were to find analogues to typical periglacial environments, temperate glacial ice could be carefully compared to periglacial "seasonally active permafrost" [10], where ice coexists with liquid water, or to periglacial "basal cryopeg", both being cryotic, but not completely frozen. Any glacial ice is frozen by definition; therefore, if regarded as a monomineral rock, it could be treated as permafrost. The only constituents of a glacier that can have positive temperature are water and morainic inclusions at or close to the glacier surface.

In the final aforementioned work, Dobiński et al. [6] already built on the notion that temperate glacial ice is cryotic, but not frozen. This is puzzling, because "frozen" is the past participle of the verb "to freeze", which means turned into ice [11]. In Lliboutry [12], one can read the following: "The temperature in a temperate glacier has been estimated by Harrison (1972) from the refreezing of water in a bore hole: θ = −0.03 °C." (see Harrison [13]). Harrison [13,14] clearly demonstrated that a temperate glacier body has subzero temperature. Without any doubts, freezing occurs in the cryotic body of a temperate glacier and frozen water (ice) constitutes most of its mass. Therefore, the notion that temperate glacial ice is not frozen is improper.

The glacial–periglacial interaction model of Dobiński et al. [6] was further improved by naming the inner-glacial boundary a "cold–temperate transition surface" (CTS), after Pettersson et al. [15], separating "temperate glacial ice permafrost" from "cold glacier ice permafrost" (Figure 1). However, this model is still problematic, because it includes 0 °C surface running inside the glacier, not around it.

I modified this model to show the correct placement of the 0 °C surface (Figure 2), the position of cryotic subglacial layer (rocks or sediments) and the permafrost limited to the area outside an active glacier. Placing the cryotic subglacial layer in the model is justified by the fact that basal temperate ice is at its pressure melting point [12–14] or below, which can result from the Robin heat pipe effect [16]. Therefore, meltwater escaping the basal ice into the subglacial porous layer is responsible for its cryotic state.

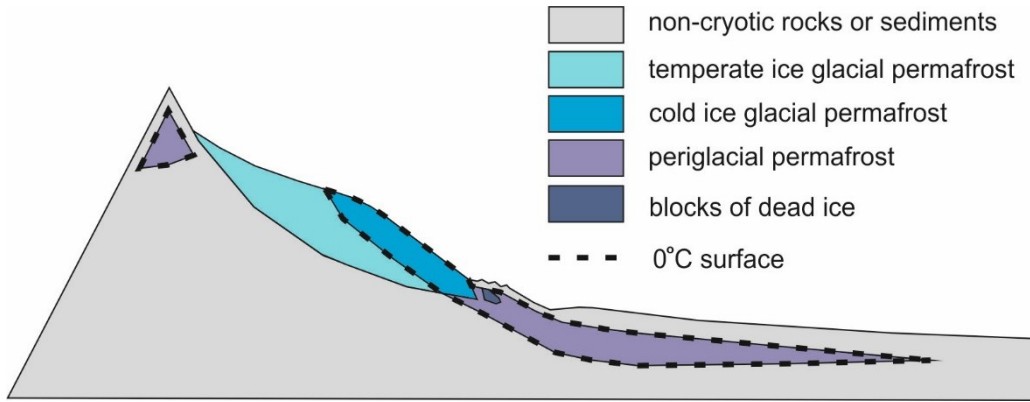

**Figure 1.** Polythermal glacier–permafrost interactions according to Dobiński et al. [6] (modified), based on previous models form Dobiński [2] and Dobiński et al. [5]; the 0 °C surface inside the glacier is called the cold–temperate transition surface (CTS).

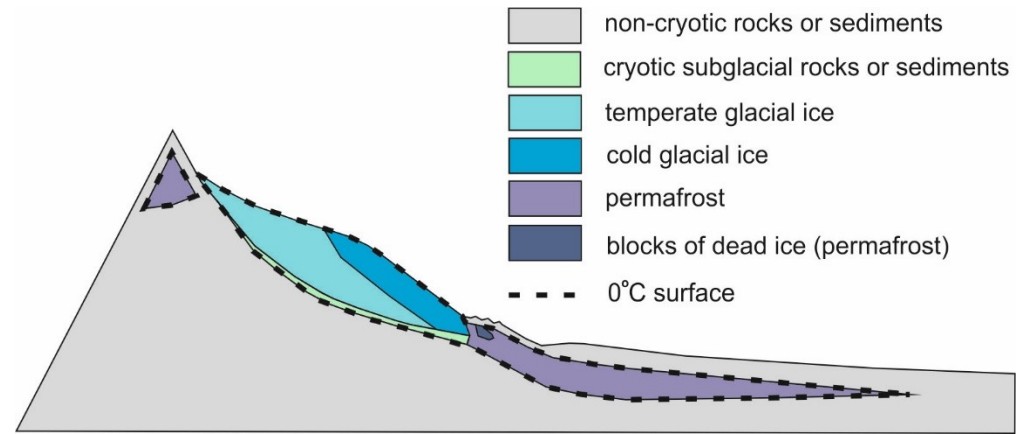

**Figure 2.** Reinterpretation of Figure 1: Polythermal glacier–permafrost interactions, based on Dobiński et al. [6], modified to include a cryotic subglacial layer and with permafrost limited to areas outside an active glacier; the thickness of the cryotic subglacial layer is not in scale as it can be very thin.

## 4. Conclusions

Works of Dobiński [2–4] and Dobiński et al. [5,6] raise important issues about the spatial relationship between the glacial and periglacial domains and rightly draw our attention to this important problem. However, I question the rationale of placing 0 °C surface inside a polythermal glacier. Two solutions are possible:

i)   limiting permafrost to a mineral ground that, according to the Cryosphere Glossary [9], excludes active glaciers but includes dead ice blocks above which an active layer (non-ice) develops;

or

ii)  broadening our understanding of permafrost to include all glaciers (cold, polythermal and temperate) but also subglacial sediments which are cryotic due to saturation with cryotic glacial meltwater under pressure.

Both solutions are correct if we accept the purely thermal definition of permafrost and that an active layer development is not a prerequisite. I am inclined towards the more traditional understanding of permafrost where active glaciers are excluded from permafrost (Figure 2), because it would clarify the issue and justify the necessity of an active layer determined above any permafrost bodies. However, it must be remembered that, in high-mountain or polar zones, glacial and periglacial domains frequently coexist and it can be difficult to draw a boundary between them [17].

The contemporary global warming issue has triggered increased scientific research in geosciences, focusing on the following types of cryotic elements of the natural environment, among others: glacial ice, snow cover, permafrost and sea ice. Proper predictions of landscape changes in the polar and high-alpine zones are possible only if we gain a good understanding of the nature of these four elements and how they individually respond to climatic changes. Extending the term "permafrost" to active glaciers (or snow) can result in a blurring of our understanding of the four cryotic elements of the Earth's system.

**Funding:** Publication financed by the Faculty of Geography and Regional Studies of the University of Warsaw.

**Acknowledgments:** I would like to thank Wojciech Dobiński for inspiration to analyze the complex issue of permafrost-glacier interactions.

**Conflicts of Interest:** The author declares no conflict of interest. The funder had no role in the design of the study; in the collection, analyses, or interpretation of data; in the writing of the manuscript, or in the decision to publish the results.

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
