# Peer review of "Should Glaciers Be Considered Permafrost?"

_geosciences, doi:10.3390/geosciences9120517_

Round 1

Reviewer 1 Report

The author of this commentary paper evaluates the concept of extending the term “permafrost” to any parts of active glaciers and refers above all to previous work by DobiÅ„ski et al. The author argues that the cryofront (0°C isoline) cannot be used as a criterion for (glacial) permafrost within permafrost-glacier interactions as was done by DobiÅ„ski et al. He advocates a spatially extended use of the cryofront but limited in his model the distribution of permafrost outside of an active glacier and suggest a traditional use of the term permafrost, which excludes active glaciers and above all defines the presence of an active layer as a prerequisite for permafrost.

I think that the article is a good commentary. The argumentation of the author is logical and understandable. Overall, I think that the commentary can be published in the present state after minor changes.

Nevertheless, my recommendation is to extend the discussion and conclusion of the article into something more than only about the thermal definition of the term permafrost, as it has been done here. Another and not unimportant aspect that could be summed up is that, especially in today's context of global warming and the goal of understanding and predicting landscape changes in the Arctic, the complex systems of glaciers and permafrost and their specific and individual responses to environmental changes should not be generalized in the use of simplified terminology.

I just found a few spelling errors:

Line 20: “The aim of this commentary is to show…”

Line 22: “…or their parts from permafrost.”

Line 48: “…, and a perennial snow patch…”

Author Response

I would like to thank very much the reviewer for the comments and linguistic adjustments.

Each of the reviewer’s critical comments (in italics) is followed by my response.

Nevertheless, my recommendation is to extend the discussion and conclusion of the article into something more than only about the thermal definition of the term permafrost, as it has been done here.

I have added a paragraph to Discussion (about the cryotic subglacial layer, lines 91-94) and one the Conclusion (see the comment below).

Another and not unimportant aspect that could be summed up is that, especially in today's context of global warming and the goal of understanding and predicting landscape changes in the Arctic, the complex systems of glaciers and permafrost and their specific and individual responses to environmental changes should not be generalized in the use of simplified terminology.

I extended Conclusion by adding a paragraph (lines 122-127) where I emphasise the need of clarity in understanding of the scientific terms (in a light of the global climate change and increased scientific interest in glaciers, snow cover, permafrost and sea ice).

Line 20: “The aim of this commentary is to show…”

Corrected accordingly

Line 22: “…or their parts from permafrost.”

Corrected accordingly

Line 48: “…, and a perennial snow patch…”

Corrected accordingly (in lines 47-48 aslo spacing was corrected)

Reviewer 2 Report

This commentary offers a critique of the idea that (parts of) glaciers may be considered "permafrost" based on the thermal definition (material less than 0C for at least 2 years), as proposed by W. Dobinski. The primary criticism is that this interpretation leads to misconceptions, especially in the case of active glaciers and in the distinction between frozen vs. cryotic material. The author provides a re-interpretation of Dobinski's interactions between glaciers and permafrost, and concludes by advocating for the traditional definition of permafrost (which excludes glaciers).

General comments:

Abstract wording was a little vague; it was not entirely clear at first read-through that the article was a critique of the idea that glaciers should be considered permafrost. Including a clause somewhere in the abstract (…"it has been proposed that [discussion of proposed hypothesis]… but here [statement indicating that this is a critique]…" would emphasize the intention of the commentary.

The referencing of the sequence of Dobinski papers discussing glacial permafrost are hard to follow – consider also including the years of these publications in the discussion, perhaps in the form of a defined shorthand (i.e., "Dobinski [2]" to "D-2006 [2]") to make the timeline readily-apparent so the reader does not need to continually go between the text and the citation list.

Perhaps beyond the scope of this work, but where would dry permafrost fit into this discussion? It would not be "frozen" in the sense of the implications for the presence of ice, but would clearly meet the thermal/cryotic definition. In addition, if there were surface material that never reaches a temperature above 0C (i.e., no active layer – unlikely on Earth today, but prominent elsewhere in the solar system), how would that fit into the existing definitions of permafrost? These nuances may ultimately help to clarify this debate.

Specific comments:

Line 8: "Expect" to "except"

Line 16: "first" instead of "for the first time"

Line 20: "…this commentary IS", not "it"

Line 22: "form" to "from"

Line 25: Replace "tells" with a different phrasing; i.e., "permafrost is defined as…" or "The general definition is…" followed by the text of the definition in quotes would be clearer.

Line 28: "…which brought him to promote the idea…" is very wordy. Consider rephrasing as: "…far reaching conclusions: the idea of glacial permafrost…"

Line 32: Recommend using "would" instead of "should" here.

Line 50: "Misconceptions" misspelled

Line 87: I don't see the distinction between "temperate glacial ice permafrost" and "temperate glacier ice permafrost" (should one of these be "cold ice glacial permafrost"?)

Lines 90-91 (and Figure 2): I think additional discussion is warranted here, including the justification behind the use of terms (e.g., would "cryotic subglacial sediments" be considered permafrost under either definition?). It appears that you are entirely rejecting Dobinski's hypotheses regarding glacial permafrost and demonstrating your conclusion (i) in this figure prior to the introduction of your conclusions. Perhaps modifying the legend to demonstrate classifications from both the traditional permafrost definition AND the Dobinski interpretation that applies a cryotic definition for permafrost would illuminate the distinction between the two (i.e., "cold glacial ice OR glacial permafrost")? In addition, the provenance of Figure 2 should be clearer in the captions (i.e., "Figure 2: Reinterpretation of Figure 1, to account for…").

Author Response

I would like to thank very much the reviewer for the comments and linguistic adjustments.

Each of the reviewer’s critical comments (in italics) is followed by my response.

Abstract wording was a little vague; it was not entirely clear at first read-through that the article was a critique of the idea that glaciers should be considered permafrost. Including a clause somewhere in the abstract (…"it has been proposed that [discussion of proposed hypothesis]… but here [statement indicating that this is a critique]…" would emphasize the intention of the commentary.

I disagree with this comment, because the very first sentence is: “This commentary critically evaluates concepts of extending permafrost to any parts of an active glacier.” Therefore, it should be clear that this is a critique and that I disagree with the notion of including glaciers (any part of them!) into permafrost.

The referencing of the sequence of Dobinski papers discussing glacial permafrost are hard to follow – consider also including the years of these publications in the discussion, perhaps in the form of a defined shorthand (i.e., "Dobinski [2]" to "D-2006 [2]") to make the timeline readily-apparent so the reader does not need to continually go between the text and the citation list.

I am not sure, if replacing “Dobinski [2]" to "D-2006 [2]" is in accordance with the journal rules of citing. There are five articles of DobiÅ„ski and they are numbered form 2 to 6 following the timeline (chronology of publication), so it is a logical sequence. I am afraid that changes introduced would only make the reader more puzzled or would uselessly occupy space.

Perhaps beyond the scope of this work, but where would dry permafrost fit into this discussion? It would not be "frozen" in the sense of the implications for the presence of ice, but would clearly meet the thermal/cryotic definition.

Dry cruotic ground (e.g. a dune) is permafrost according to IPA definition, but not NSIDC definition which requires freezeng. However, my polemic targets a notion that glaciers a permafrost, and glaciers have nothing to do with dry permafrost. I would rather not go into further discussion of every aspect of permafrost issues. Nevertheless, in the original version of my manuscript (and now, lines 37-38) I shortly address the dry permafrost case.

In addition, if there were surface material that never reaches a temperature above 0C (i.e., no active layer – unlikely on Earth today, but prominent elsewhere in the solar system), how would that fit into the existing definitions of permafrost? These nuances may ultimately help to clarify this debate.

I address this issue in Conclusion where I write that it is in accordance with the purely thermal definition if we agree to : “broadening our understanding of permafrost to include all glaciers (cold, polythermal and temperate)….”.  This would also apply to a superficial mineral (non-ice) ground layer on any planet. However, taken the reality of Earth’s environment, a distinction between a glacier (temperate, cold or polythermal) and traditional permafrost is needed. If I extend my discussion into other celestial bodies, I will probably trigger further comments by reviewers, which would make my polemical article distracted from the issue of “glacial permafrost”.

Line 8: "Expect" to "except"

Corrected accordingly

Line 16: "first" instead of "for the first time"

Corrected accordingly

Line 20: "…this commentary IS", not "it"

Corrected accordingly

Line 22: "form" to "from"

Corrected accordingly

Line 25: Replace "tells" with a different phrasing; i.e., "permafrost is defined as…" or "The general definition is…" followed by the text of the definition in quotes would be clearer.

Now, this sentence is (lines 24-25): “According to the International Permafrost Association (IPA), permafrost is defined as a ground remaining at or below 0oC for at least two consecutive years [7,8].”

Line 28: "…which brought him to promote the idea…" is very wordy. Consider rephrasing as: "…far reaching conclusions: the idea of glacial permafrost…"

I would rather maintain my phrasing in order to emphasise that Dobiński advocates/promotes his idea (in his articles and on conferences). The suggestion of the Reviewer changes the meaning of this sentence so that it would no longer convey my message..

Line 32: Recommend using "would" instead of "should" here.

Corrected accordingly

Line 50: "Misconceptions" misspelled

Corrected accordingly

Line 87: I don't see the distinction between "temperate glacial ice permafrost" and "temperate glacier ice permafrost" (should one of these be "cold ice glacial permafrost"?)

Of course, now it is: separating “temperate glacial ice permafrost” from “cold glacier ice permafrost” (now line 86).

Lines 90-91 (and Figure 2): I think additional discussion is warranted here, including the justification behind the use of terms (e.g., would "cryotic subglacial sediments" be considered permafrost under either definition?).

I have added a short justification of the cryotic subglacial layer, which resulted in adding one extra reference (nr 17: Robin 1976). lines 91-94. I have also slightly modified Fig 2 (extended the light-green subglacial layer to the very front of the glacier and change it’s description into “cryotic subglacial rocks or sediments”.

It appears that you are entirely rejecting Dobinski's hypotheses regarding glacial permafrost and demonstrating your conclusion (i) in this figure prior to the introduction of your conclusions. Perhaps modifying the legend to demonstrate classifications from both the traditional permafrost definition AND the Dobinski interpretation that applies a cryotic definition for permafrost would illuminate the distinction between the two (i.e., "cold glacial ice OR glacial permafrost")?

I have modified the legend of Fig 2: “cryotic subglacial sediments” is changed into “cryotic subglacial rocks and sediments”, but see no reason for further modifications. Further extension of the legend would complicate the message conveyed in this figure.

In addition, the provenance of Figure 2 should be clearer in the captions (i.e., "Figure 2: Reinterpretation of Figure 1, to account for…").

I modified the Fig. 2 caption to comply with your suggestion and to indicate that the thickness of the cryotic subglacial layer is not is scale (lines 100-103).